# Evolving scientific discovery by unifying data and background knowledge with AI Hilbert

Ryan Cory-Wright [1] ✉, Cristina Cornelio [2], Sanjeeb Dash[3], Bachir El Khadir[3] & Lior Horesh [3]

The discovery of scientific formulae that parsimoniously explain natural phenomena and align with existing background theory is a key goal in science. Historically, scientists have derived natural laws by manipulating equations based on existing knowledge, forming new equations, and verifying them experimentally. However, this does not include experimental data within the discovery process, which may be inefficient. We propose a solution to this problem when all axioms and scientific laws are expressible as polynomials and argue our approach is widely applicable. We model notions of minimal complexity using binary variables and logical constraints, solve polynomial optimization problems via mixed-integer linear or semidefinite optimization, and prove the validity of our scientific discoveries in a principled manner using Positivstellensatz certificates. We demonstrate that some famous scientific laws, including Kepler's Law of Planetary Motion and the Radiated Gravitational Wave Power equation, can be derived in a principled manner from axioms and experimental data.

A fundamental problem in science involves explaining natural phenomena in a manner consistent with noisy experimental data and a body of potentially inexact and incomplete background knowledge about the universe's laws[1]. In the past few centuries, The Scientific Method[2] has led to significant progress in discovering new laws. Unfortunately, the rate of emergence of these laws and their contribution to economic growth is stagnating relative to the amount of capital invested in deducing them[3,4]. Indeed, Dirac[5] noted that it is now more challenging for first-rate physicists to make second-rate discoveries than it was previously for second-rate physicists to make first-rate ones, while Arora et al.[6] found that the marginal value of scientific discoveries to large companies has declined since the fall of the Berlin Wall. Moreover, Bloom et al.[7] have found that research productivity in the United States halves every thirteen years because good scientific ideas are getting harder to find. This phenomenon can be partly explained by analogy to the work of Cowen[8]. Namely, The Scientific Method has picked most of the low-hanging fruit in science, such as natural laws that relate physical quantities using a small number of low-degree polynomials. This calls for more disciplined and principled

alternatives to The Scientific Method, which integrates background information and experimental data to generate and verify higher dimensional laws of nature, thereby promoting scientific discovery (c.f. refs. 9, 10). Accordingly, Fig. 1 provides an overview of scientific discovery paradigms.

Even as the rate of scientific discovery has decreased, the scalability of global optimization methods has significantly improved. Indeed, as we argue in this paper, global optimization methods are now a mature technology capable of searching over the space of scientific laws - owing to Moore's law and significant theoretical and computational advances by the optimization community see refs. 11–13, for reviews. For instance, Bertsimas and Dunn[14], Chap. 1 observed that the speedup in raw computing power between 1991 and 2015 is at least six orders of magnitude. Additionally, polynomial optimization has become much more scalable since the works of Lasserre[15] and Parrilo[16], and primal-dual interior-point methods[17–19] have improved considerably, with excellent implementations now available in, for example, the `Mosek` solver[20]. Even methods for non-convex quadratically constrained problems have already achieved

[1]Department of Analytics, Marketing and Operations, Imperial College Business School, London, UK. [2]Samsung AI, Cambridge, UK. [3]IBM Thomas J. Watson Research Center, Yorktown Heights, USA. ✉e-mail: r.cory-wright@imperial.ac.uk

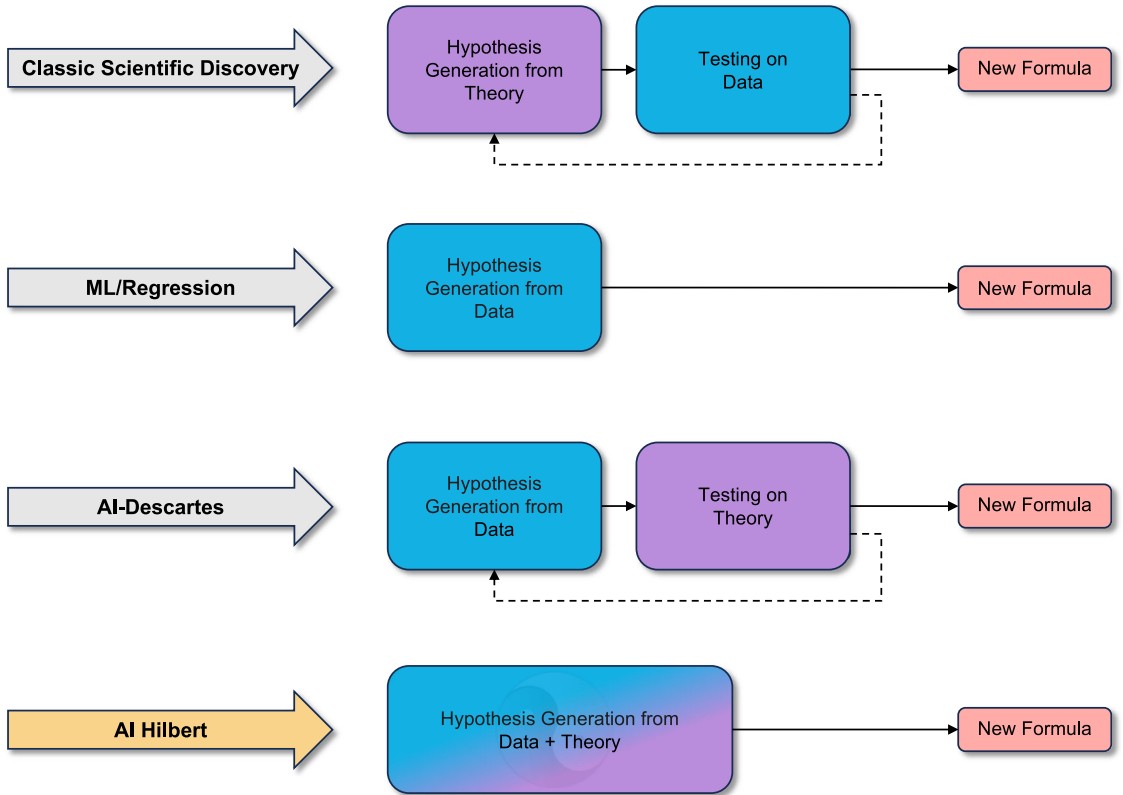

**Fig. 1 | Comparison of scientific discovery paradigms.** Traditional scientific discovery formulates hypotheses using existing theory and observed phenomena. These hypotheses are validated and tested using data. In contrast, machine learning methods rely on large datasets to identify patterns. AI-Descartes[43] proposes an inversion of the traditional scientific discovery paradigm. It generates hypotheses from observed data and validates them against known theories. However, in AI-Descartes, theory and data remain disjoint and do not mutually enhance one another. In contrast, our work, `AI-Hilbert`, combines data and theory to formulate hypotheses. Unlike conventional methods, insights into data and knowledge embedded in the theory collaboratively reduce the search space. These two components complement each other: theory compensates for noisy or sparse data, while data compensates for inconsistent or incomplete theory. Note that blue denotes components associated with data, purple denotes components linked to theory, and dashed lines represent iterative processes.

machine-independent speedups of nearly 200 since their integration within commercial solvers in 2019[12,21].

In this paper (we introduce our notation in our "Methods" section, code and data used for this work are available at ai-hilbert.github.io), we propose a new approach to scientific discovery that leverages these advances by the optimization community. Given a set of background axioms, theorems, and laws expressible as a basic semi-algebraic set (i.e., a system of polynomial equalities and inequalities) and observations from experimental data, we derive new laws representable as polynomial expressions that are either exactly or approximately consistent with existing laws and experimental data by solving polynomial optimization problems via linear and semidefinite optimization. By leveraging fundamental results from real algebraic geometry, we obtain formal proofs of the correctness of our laws as a byproduct of the optimization problems. This is notable, because existing automated approaches to scientific discovery[22–25], as reviewed in Section 1 of our supplementary material, often rely upon deep learning techniques that do not provide formal proofs and are prone to hallucinating incorrect scientific laws that cannot be automatically proven or disproven, analogously to output from state-of-the-art Large Language Models such as GPT-4[26]. As such, any new laws derived from these systems cannot easily be explained or justified.

Conversely, our approach discovers new scientific laws by solving an optimization problem to minimize a weighted sum of discrepancies between the proposed law and data, plus the distance between the proposed law and its projection onto the set of symbolic laws derivable from background theory. As a result, our approach discovers scientific laws alongside proof of their consistency with existing background

theory by default. Moreover, our approach is scalable; it runs in polynomial time with respect to the number of symbolic variables and axioms (when the degree of the polynomial certificates we search over is bounded) with a complete and consistent background theory.

We believe our approach could be a first step towards discovering new laws of the universe involving higher-degree polynomials, which are impractical for scientists to discover without the aid of modern optimization solvers and high-performance computing environments. Further, our approach is potentially useful for reconciling mutually inconsistent axioms. Indeed, if a system of scientific laws is mutually inconsistent (in the sense that no point satisfies all laws simultaneously), our polynomial optimization problem offers a formal proof of its inconsistency.

## Results

We propose a new paradigm for scientific discovery that derives polynomial laws simultaneously consistent with experimental data and a body of background knowledge expressible as polynomial equalities and inequalities. We term our approach `AI-Hilbert`, inspired by the work of David Hilbert, one of the first mathematicians to investigate the relationship between sum-of-squares and non-negative polynomials[27].

Our approach automatically provides an axiomatic derivation of the correctness of a discovered scientific law, conditional on the correctness of our background theory. Moreover, in instances with inconsistent background theory, our approach can successfully identify the sources of inconsistency by performing best subset selection to determine the axioms that best explain the data. This is notably different from current data-driven approaches to scientific discovery,

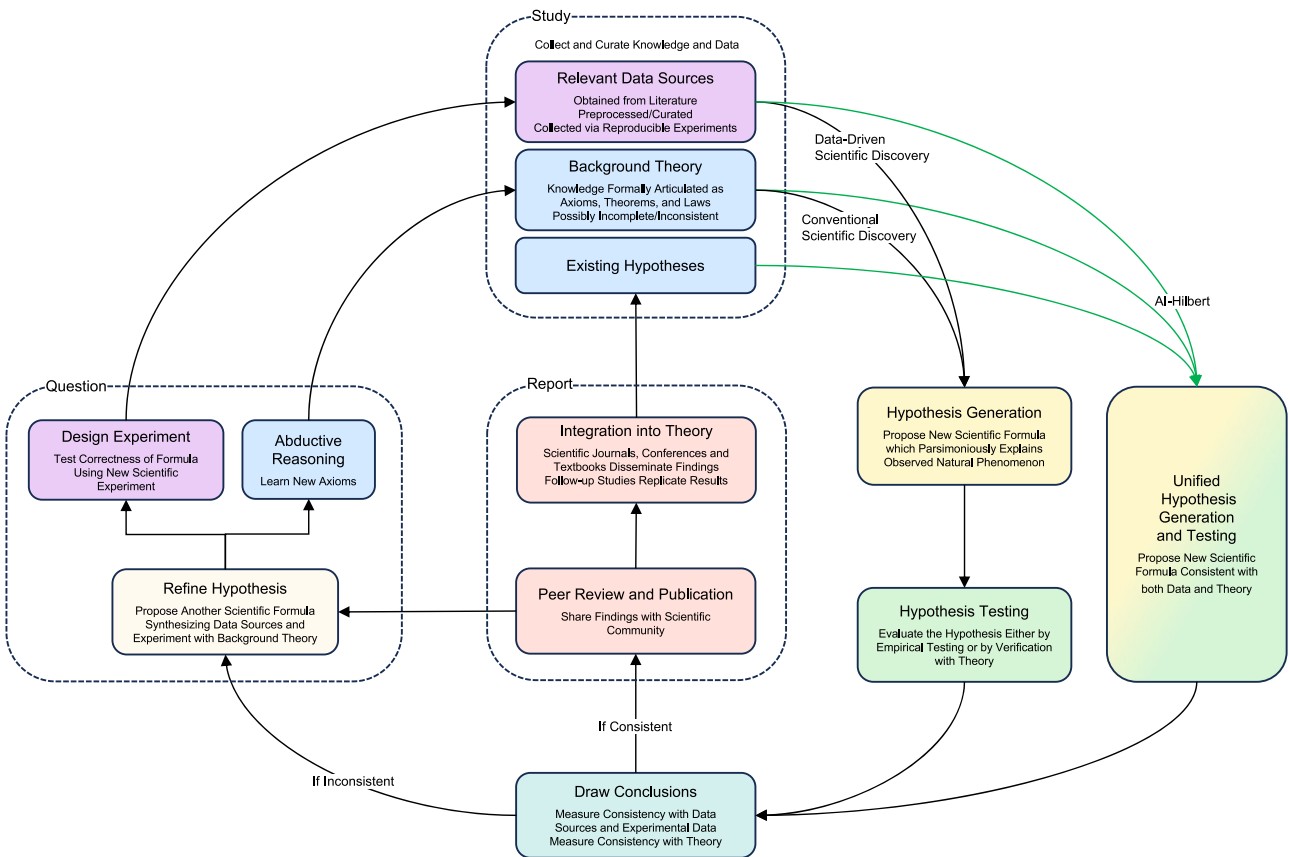

**Fig. 2 | The scientific method with scientific discoveries made via classical methods, data-driven methods, or AI-Hilbert.** AI-Hilbert proposes scientific laws consistent with a body of background theory formally articulated as polynomial equalities, inequalities, and relevant data sources. This likely allows scientific discoveries to be made using fewer data points than state-of-the-art approaches, and for missing scientific axioms to be deduced via abductive reasoning as part of the scientific discovery process. On the other hand, existing approaches to scientific discovery propose laws that may be inconsistent with either background theory or existing data sources. Note that blue denotes components associated with logical theories, purple denotes components linked to data, yellow denotes hypothesis generation, green denotes hypothesis testing, orange denotes discovery reporting, white denotes hypothesis refinement, teal denotes evaluation of the discovery, dashed lines represent macro components and green lines represent the input of AI-Hilbert.

which often generate spurious laws in limited data settings and fail to differentiate between valid and invalid discoveries or provide explanations of their derivations. We illustrate our approach by axiomatically deriving some of the most frequently cited natural laws in the scientific literature, including Kepler's Third Law and Einstein's Relativistic Time Dilation Law, among other scientific discoveries.

A second contribution of our approach is that it permits fine-grained control of the tractability of the scientific discovery process, by bounding the degree of the coefficients in the Positivstellensatz certificates that are searched over (see our "Methods" section for a formal statement of the Positivstellensatz). This differs from prior works on automated scientific discovery, which offers more limited control over their time complexity. For instance, in the special case of scientific discovery with a complete body of background theory and no experimental data, to our knowledge, the only current alternative to our approach is symbolic regression see, e.g., ref. [28], which requires genetic programming or mixed-integer nonlinear programming techniques that are not guaranteed to run in polynomial time. On the other hand, our approach searches for polynomial certificates of a bounded degree via a fixed level of the sum-of-squares hierarchy[15,16], which can be searched over in polynomial time under some mild regularity conditions[17,29].

To contrast our approach with existing approaches to scientific discovery, Fig. 2 depicts a stylized version of the scientific method. In this version, new laws of nature are proposed from background theory (which may be written down by humans, automatically

extracted from existing literature, or even generated using AI[30]) and experimental data, using classical discovery techniques, data-driven techniques, or AI-Hilbert. Observe that data-driven discoveries may be inconsistent with background theory, and discoveries via classical methods may not be consistent with relevant data sources, while discoveries made via AI-Hilbert are consistent with background theory and relevant data sources. This suggests that AI-Hilbert could be a first step toward discovery frameworks that are less likely to make false discoveries. Moreover, as mentioned in the introduction, AI-Hilbert uses background theory to restrict the effective dimension of the set of possible scientific laws, and, therefore requires less data to make scientific discoveries than purely data-driven approaches.

## Scientific discovery as polynomial optimization

Our scientific discovery method (AI-Hilbert) aims to discover an unknown polynomial formula $q(\cdot) \in \mathbb{R}[x]$ which describes a physical phenomenon, and is both consistent with a background theory of polynomial equalities and inequalities $\mathcal{B}$ (a set of axioms) and a collection of experimental data $\mathcal{D}$ (defined below). We provide a high-level overview of AI-Hilbert in Fig. 3 and summarize our procedure in Algorithm 1. The inputs to AI-Hilbert are a four-tuple $(\mathcal{B}, \mathcal{D}, \mathcal{C}(\Lambda), d^c)$, where:

- $\mathcal{B}$ denotes the relevant background theory, expressed as a collection of axioms in the scientific discovery setting, i.e., the polynomial laws relevant for discovering $q$. It is the union of the

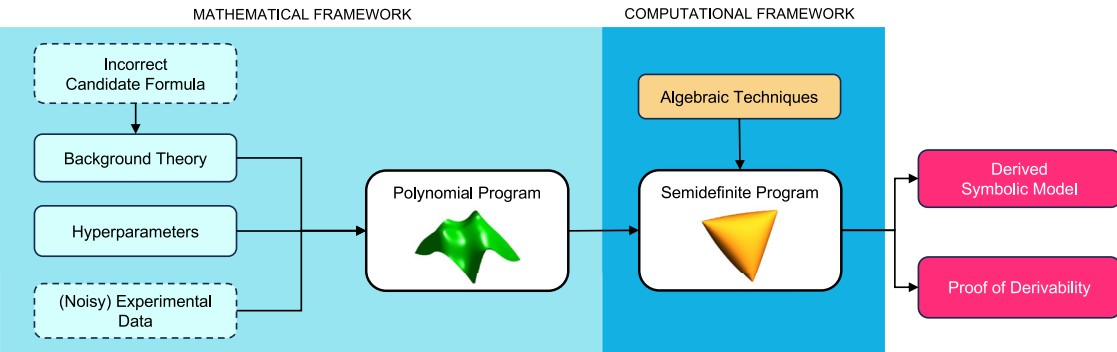

**Fig. 3 | Schematic illustration of AI-Hilbert and its components.** Using background knowledge encoded as multivariate polynomials, experimental data, and hyperparameters (e.g., a sparsity constraint on the background theory) to control our model's complexity, we formulate scientific discovery as a polynomial optimization problem, reformulate it as a semidefinite optimization problem, and solve it to obtain both a symbolic model and its formal derivation. Dashed boxes correspond to optional components. An example is introducing an incorrect candidate formula as a new axiom in the background theory.

inequalities $\{g_1(\boldsymbol{x}) \geq 0, ..., g_k(\boldsymbol{x}) \geq 0\}$ defining $\mathcal{G}$ and the equalities $\{h_1(\boldsymbol{x}) = 0, ..., h_l(\boldsymbol{x}) = 0\}$ defining $\mathcal{H}$. Further, $\mathcal{B}$ is defined over $n$ variables $x_1, ..., x_n$. However, only $t$ of these $n$ variables can be measured and are directly relevant for explaining the observed phenomenon. In particular, we let $x_1$ denote the dependent variable. The remaining $n - t$ variables appear in the background theory but are not directly observable. The background theory $\mathcal{B}$ is defined as *complete* if it contains all the axioms necessary to formally prove the target formula, *incomplete* otherwise. Moreover, $\mathcal{B}$ is called *inconsistent* if it contains axioms that contradict each other, *consistent* otherwise (we define these terms more rigorously in our "Methods" section). One might also exclude measurements for certain variables $x_j$ if they are trivially connected to other measurable variables. For instance, the period of rotation can trivially be obtained from the frequency of rotation, and it is better to avoid having both in the list of measured variables in our system. We treat known quantities such as $c$ - the speed of light - as measurable entities.

- $\mathcal{D} := \{\bar{\boldsymbol{x}}_i\}_{i=1}^m$ denotes a collection of data points, or measurements of an observed physical phenomenon, which may be contaminated by noise, e.g., from measurement error. We assume that $\bar{\boldsymbol{x}}_i \in \mathbb{R}^n$ and $\bar{x}_{i,j} = 0$ for $j \geq t + 1$, i.e., the value of $\bar{x}_{i,j}$ – the $j$th entry of $\bar{\boldsymbol{x}}_i$ – is set to zero for all variables $j$ that cannot or should not be measured.

- $\mathcal{C}$ denotes a set of constraints and bounds which depend on a set of hyper-parameters $\Lambda$. Specifically, we consider a global bound on the degree of the polynomial $q$; a vector $\boldsymbol{d}$ restricting individual variable degrees in $q$; a hyperparameter $\lambda$ that models the fidelity to background theory and data; and a bound over the number of axioms that should be included in a formula derivation.

- $d^c(\cdot, \mathcal{G} \cap \mathcal{H})$ denotes a distance function that defines the distance from an arbitrary polynomial to the background theory. We formally define $d^c$ in our "Methods" section.

Algorithm 1 provides a high-level description of AI-Hilbert. The procedure first combines the background theory $\mathcal{B}$ and data $\mathcal{D}$ to generate a polynomial optimization problem $\mathbb{Pr}$ which targets a specific concept identified by a dependent - or target - variable included in the set of observable entities that can be measured in the environment $(x_1, ..., x_t)$. This is achieved by leveraging the distance $d^c$ (formally defined in our "Methods" section) and integrating the bounds and constraints $\mathcal{C}$ (with their hyperparameters $\Lambda$) via the `PolyJuMP.jl` Julia package[31]. This corresponds to the `Formulate` step of Algorithm 1, which we formalize in our "Methods" section.

**Algorithm 1.** `AI-Hilbert` for Scientific Discovery

> **Input:** $(\mathcal{B}, \mathcal{D}, \mathcal{C}(\Lambda), d^c)$
> 1: $\mathbb{Pr} \leftarrow$ `Formulate`$(\mathcal{B}, \mathcal{D}, \mathcal{C}(\Lambda), d^c)$
> 2: $\mathbb{Pr}^{sd} \leftarrow$ `Reduce`$(\mathbb{Pr})$
> 3: $q(\boldsymbol{x}) \leftarrow$ `Solve`$(\mathbb{Pr}^{sd})$
> **Output:** $q(\boldsymbol{x}) = 0$
> **Output:** $\boldsymbol{\alpha}, \boldsymbol{\beta}$

`AI-Hilbert` then reformulates the problem $\mathbb{Pr}$ as a semidefinite (or linear if no inequalities are present in the background theory) optimization problem $\mathbb{Pr}^{sd}$, by leveraging standard techniques from sum-of-squares optimization that are now integrated within the `SumOfSquares.jl` and `PolyJuMP.jl` Julia packages. This corresponds to the `Reduce` step of Algorithm 1.

Next, `AI-Hilbert` solves $\mathbb{Pr}^{sd}$ using a mixed-integer conic optimization solver such as `Gurobi`[21] or `Mosek`[20]. This corresponds to the `Solve` step of Algorithm 1.

`AI-Hilbert` then outputs a candidate formula of the form $q(\boldsymbol{x}) = 0$ where the only monomials with nonzero coefficients are those that only contain the variables $x_1, ..., x_t$, the independent and dependent variables that are observed in the environment. The background theory may contain additional variables $x_{t+1}, ..., x_n$ that are not observed in the environment and that will not appear in the derived law. This is because the axioms in the background theory are not constraints on the functional form of the target polynomial, but rather general scientific laws describing the environment, often not including any of the quantities/variables observed in the data.

Finally, `AI-Hilbert` returns polynomial multipliers $\{\alpha_i\}_{i=0}^k, \{\beta_j\}_{j=1}^l$ (sums-of-squares polynomials and arbitrary polynomials, respectively) such that

$$q(\boldsymbol{x}) = \alpha_0(\boldsymbol{x}) + \sum_{i=1}^k \boldsymbol{\alpha}_i(\boldsymbol{x}) g_i(\boldsymbol{x}) + \sum_{j=1}^l \boldsymbol{\beta}_j(\boldsymbol{x}) h_j(\boldsymbol{x}) \qquad (1)$$

if $d^c(q, \mathcal{G} \cap \mathcal{H}) = 0$, which is a certificate of the fact that $q$ is derivable from the background theory. If $d^c > 0$, then `AI-Hilbert` returns a certificate that $q$ is approximately derivable from the background theory, and $q$ is approximately equal to $\alpha_0(\boldsymbol{x}) + \sum_{i=1}^k \boldsymbol{\alpha}_i(\boldsymbol{x}) g_i(\boldsymbol{x}) + \sum_{j=1}^l \boldsymbol{\beta}_j(\boldsymbol{x}) h_j(\boldsymbol{x})$.

**Notion of distance to background theory and model complexity**

In scientific discovery settings, scientists often start with experimental measurements and a set of polynomial equalities and inequalities (axioms) that they believe to be true. From these axioms and measurements, they aim to deduce a new law, explaining their

data, which includes one or more dependent variables and excludes certain variables. The simplest case of scientific discovery involves a consistent and correct set of axioms that fully characterize the problem. In this case, the Positivstellensatz (see "Methods" section) facilitates the discovery of new scientific laws via deductive reasoning, without using any experimental data. Indeed, under an Archimedean assumption (see Theorem 1 in our "Methods" section), the set of all valid scientific laws corresponds precisely to the pre-prime (see ref. 32 for a definition) generated by our axioms[33] and searching for the simplest polynomial version of a law that features a given dependent variable corresponds to solving an easy linear or semi-definite problem.

Unfortunately, it is not uncommon to have a set of axioms that is inconsistent (meaning that there are no values of $x \in \mathbb{R}^n$ that satisfy all laws simultaneously) or incomplete (meaning the axioms do not 'span' the space of all derivable polynomials). Therefore, AI-Hilbert requires a notion of a distance between a body of background theory (which, in our case, consists of a set of polynomial equalities and inequalities) and a polynomial. We now establish this definition, treating the inconsistent and incomplete cases separately; note that the inconsistent and incomplete cases may be treated via the inconsistent case. We remark that refs. 34,35 propose related notions of the distance between (a) a point and a variety defined by a set of equality constraints and (b) the distance between two semialgebraic sets via their Hausdorff distance. To our knowledge, the distance metrics in this paper are new.

**Incomplete background knowledge case.** Suppose $\mathcal{B}$ is a background theory (consisting of equalities and inequalities in $\mathcal{G}$ and $\mathcal{H}$), and the axioms are not inconsistent, meaning that $\mathcal{G} \cap \mathcal{H} \neq \emptyset$. Then, a natural notion of distance is the $\ell_2$ coefficient distance $d^c$ between $q$ and $\mathcal{G} \cap \mathcal{H}$, which is given by:

$$d^c(q, \mathcal{G} \cap \mathcal{H}) := \min_{\substack{\alpha_0, \ldots, \alpha_k \in \Sigma[\boldsymbol{x}]_{n,2d}, \\ \beta_1, \ldots, \beta_l \in \mathbb{R}[\boldsymbol{x}]_{n,2d}}} \left\| q - \alpha_0 - \sum_{i=1}^{k} \alpha_i g_i - \sum_{j=1}^{l} \beta_j h_j \right\|_2. \quad (2)$$

It follows directly from Putinar's Positivstellensatz that $d(q, \mathcal{G} \cap \mathcal{H}) = 0$ if and only if $q$ is derivable from $\mathcal{B}$. We remark that this distance has a geometric interpretation as the distance between a polynomial $q$ and its projection onto the algebraic variety generated by $\mathcal{G} \cap \mathcal{H}$. Moreover, by norm equivalence, this is equivalent to the Hausdorff distance[35] between $q$ and $\mathcal{G} \cap \mathcal{H}$.

With the above definition of $d^c$, and the fact that $\mathcal{G} \cap \mathcal{H} \neq \emptyset$, we say that $\mathcal{G} \cap \mathcal{H}$ is an incomplete set of axioms if there does not exist a polynomial $p$ with a non-zero coefficient on at least one monomial involving a dependent variable, such that $d^c(q, \mathcal{G} \cap \mathcal{H}) = 0$.

**Inconsistent background knowledge case.** Suppose $\mathcal{B}$ is an inconsistent background theory i.e, $\mathcal{G} \cap \mathcal{H} = \emptyset$. Then, a natural approach to scientific discovery is to assume that a subset of the axioms are valid laws, while the remaining axioms are scientifically invalid (or invalid in a specific context, e.g., micro vs. macro-scale). In line with the sparse regression literature (c.f. ref. 36 and related work on discovering nonlinear dynamics[37,38]), we assume that scientific discoveries can be made using at most $k$ correct scientific laws and define the distance between the scientific law and the problem data via a best subset selection problem. Specifically, we introduce binary variables $z_i$ and $y_j$ to denote whether the $i$th and $j$-th laws are consistent and require that $\alpha_i = 0$ if $z_i = 0$ and $\beta_j = 0$ if $y_j = 0$ and $\sum_i z_i + \sum_j y_j \leq \tau$ for a sparsity budget $\tau$. Furthermore, we allow a non-zero $\ell_2$ distance between the scientific law $f$ and the reduced background theory, but penalize this distance in the objective. This gives the following notion of distance between a

scientific law $q$ and the space $\mathcal{G} \cap \mathcal{H}$:

$$d^c(q, \mathcal{G} \cap \mathcal{H}) := \min \quad \left\| q - \alpha_0 - \sum_{i=1}^{k} \alpha_i g_i - \sum_{j=1}^{l} \beta_j h_j \right\|_2,$$
$$\text{s.t.} \quad \alpha_i = 0 \text{ if } z_i = 0, \forall i \in \{0, \ldots, k\},$$
$$\beta_j = 0 \text{ if } y_j = 0, \forall j \in \{1, \ldots l\},$$
$$\sum_{i=0}^{k} z_i + \sum_{j=1}^{l} y_j \leq \tau,$$
$$z_0 \ldots z_k \in \{0, 1\}, y_1, \ldots y_l \in \{0, 1\},$$
$$\alpha_0, \ldots, \alpha_k \in \Sigma[\boldsymbol{x}]_{n,2d}, \beta_1, \ldots, \beta_l \in \mathbb{R}[\boldsymbol{x}]_{n,2d}.$$
$$(3)$$

It follows directly from the Positivstellensatz (Theorem 1 in our "Methods" section) that $d = 0$ if and only if $q$ can be derived from $\mathcal{B}$. If $\tau = m + l$, then we certainly have $d^c = 0$ since the overall system of polynomials is inconsistent and the sum-of-squares proof system can deduce that "$-1 \geq 0$" from inconsistent proof systems, from which it can claim a distance of 0. However, by treating $\tau$ as a hyper-parameter and including the quality of the law on experimental data as part of the optimization problem, scientific discoveries can be made from inconsistent axioms by incentivizing solvers to set $z_i = 0$ for inconsistent axioms $i$.

Alternatively, a practitioner may wish to explore the Pareto frontier of scientific discoveries that arise as we vary $\tau$, to detect how large the set of correct background knowledge is. Provided there is a sufficiently high penalty cost on poorly explaining scientific data via the derived law, our optimization problem prefers a subset of correct axioms with a non-zero distance $d^c$ to the derived polynomial over a set of inconsistent axioms which gives a distance $d^c = 0$.

## Trade-off between data and theory

There is a fundamental trade-off between the amount of background theory available and the amount of data needed for scientific discovery. Indeed, with a complete and consistent set of background theories, it is sometimes possible to perform scientific discovery without any experimental data via the Positivstellensatz (see "Methods" section, for a discussion). On the other hand, the purely data-driven approaches to scientific discovery reviewed in the introduction and Section 1 of the supplementary material often require many noiseless or low-noise experimental observations to successfully perform scientific discovery. More generally, as the amount of relevant background theory increases, the amount of experimental data required by AI-Hilbert to perform scientific discovery cannot increase because increasing the amount of background theory decreases the effective VC-dimension of our scientific discovery problem (see, e.g., ref. 39). This can be seen in the machine learning examples discussed in Section 4 of the supplementary material, where imposing a sparsity constraint (i.e., providing a relevant axiom) reduces the number of data observations needed to discover a ground truth model; see also ref. 40, 41 for an analysis of the asymptotic consistency of shape-constrained regression. We provide further evidence of this experimentally in our "Methods" section.

The above observations can also be explained via real algebraic geometry: requiring that a scientific law is consistent with an axiom is equivalent to restricting the space of valid scientific laws to a subset of the space of discoverable scientific laws. As such, an axiom is equivalent to infinitely many data observations that penalize scientific laws outside a subspace but provide no information that discriminates between scientific laws within the subspace.

We provide examples from the machine learning literature that illustrate this trade-off between data and theory in Section 4 of the supplementary material.

### Experimental validation

We perform a variety of experiments with different datasets to evaluate the performance of `AI-Hilbert`. We first demonstrate that `AI-Hilbert` can in some cases, obtain the desired symbolic expression purely from a complete and consistent background theory without the use of numerical data; these include the Hagen-Poisseuille equation, Radiated gravitational wave power formula, and Einstein's relativistic time dilation law. In all these cases, the number of observable variables is a strict subset of all the variables in the background theory. We next demonstrate the ability to deal with inconsistent background theory axioms and find a subset of axioms that are consistent and yield the desired symbolic expression. We also study problems where both background theory axioms and data are needed to derive the correct symbolic expression; this is the case when one does not have a complete set of background theory axioms. Finally, we compare `AI-Hilbert` against other methods that are purely data-driven (AI Feynman, PySR, Bayesian Machine Scientist) and methods that use background theory to filter out expressions derived from data (AI Descartes) and show that `AI-Hilbert` has the best performance on a test set of problems.

## Discussion

In this work, we propose a new approach to scientific discovery that leverages ideas from real algebraic geometry and mixed-integer optimization to discover new scientific laws from a possibly inconsistent or incomplete set of scientific axioms and noisy experimental data. This improves existing approaches to scientific discovery that typically propose plausible scientific laws from either background theory alone or data alone. Indeed, by combining data and background theory in the discovery process, we potentially allow scientific discoveries to be made in previously inhospitable regimes where there is limited data and/or background theory, and gathering data is expensive. We hope our approach serves as an exciting tool that assists the scientific community in efficiently and accurately explaining the natural world.

We now discuss the generality and complexity of `AI-Hilbert`.

### Implicit and explicit symbolic discovery

Most prior work (e.g., refs. [42–44]) aims to identify an unknown symbolic model $f \in \mathbb{R}[x]_{n,2d}$ of the form $y_i = f(x_i)$ for a set of independent variables of interest $x \in \mathbb{R}^n$ and a dependent variable $y \in \mathbb{R}$, while `AI-Hilbert` searches for an implicit polynomial function $q$ which links the dependent and independent variables. We do this for two reasons. First, many scientific formulae of practical interest admit implicit representations as polynomials, but explicit representations of the dependent variable as a polynomial function of the independent variables are not possible (c.f. ref. [45]). For instance, Kepler's third law of planetary motion has this property, as discussed in our "Methods" section. Second, as proven by Artin[46] to partially resolve Hilbert's 17th problem (c.f. ref. [47]), an arbitrary non-negative polynomial can be represented as a sum of squares of rational functions. Therefore, by multiplying by the denominator in Artin's representation[46], implicit representations of natural laws become a viable and computationally affordable search space.

We remark that the implicit representation of scientific laws as polynomials where $q(x) = 0$ introduces some degeneracy in the set of optimal polynomials derivable from (10), particularly in the presence of a correct yet overdetermined background theory. For instance, in our derivation of Kepler's Law of Planetary Motion, we eventually derive the polynomial $m_1 m_2 G p^2 = m_1 d_1 d_2^2 + m_2 d_1^2 d_2 + 2 m_2 d_1 d_2^2$. Since we have the axiom that $m_1 d_1 = m_2 d_2$, we could instead derive the (equivalent) formula $m_1 m_2 G p^2 = (m_1 + m_2) d_1 d_2 (d_1 + d_2)$. To partly break this degeneracy, we propose to either constrain the degree of the proof certificate and gradually increase it (as is done in (10)) or, (equivalently in a Lagrangian sense) include a term modeling the

complexity of our derived polynomial (e.g., $\|q\|_1$, the $L_1$-coefficient norm of $q$) in the objective.

### Complexity of scientific discovery

Observe that, if the degree of our new scientific law $q$ is fixed and the degree of the polynomial multipliers in the definition in $d^c$ is also fixed, then the optimization problems arising from our approach can be solved in polynomial time (under the real complexity model, or under the bit complexity model under some mild regularity conditions on the semidefinite problems that arise from our sum-of-squares problems; see Ramana[29]) with a consistent set of axioms (resp. non-deterministic polynomial time with an inconsistent set of axioms). This is because solving our scientific discovery problem with a fixed degree and a consistent set of axioms corresponds to solving a semidefinite optimization problem of a polynomial size, which can be solved in polynomial time (assuming that a constraint qualification such as Slater's condition holds)[17]. Moreover, although solving our scientific discovery problem with a fixed degree and an inconsistent set of axioms corresponds to solving a mixed-integer semidefinite optimization problem, which is NP-hard, recent evidence[48] shows that integer optimization problems can be solved in polynomial time with high probability. This suggests that our scientific discovery problem may also be solvable in polynomial time with high probability. However, if the degree of $q$ is unbounded then, to the best of our knowledge, no existing algorithm solves Problem (10) in polynomial time. This explains why searching for scientific laws of a fixed degree and iteratively increasing the degree of the polynomial laws searched over, in accordance with Occam's Razor, is a key aspect of our approach.

Inspired by the success of `AI-Hilbert` in rediscovering existing scientific laws, we now discuss some exciting research directions that are natural extensions of this work.

### Improving the generality of AI-Hilbert

This work proposes a symbolic discovery framework that combines background theory expressible as a system of polynomial equalities and inequalities, or that can be reformulated as such a system (e.g., in a Polar coordinate system, by substituting $x = r \cos \theta, y = r \sin \theta$ and requiring that $x^2 + y^2 = r^2$). However, many scientific discovery contexts involve background theory that cannot easily be expressed via polynomial equalities and inequalities, including differential operators, integrals, and limits, among other operators. Therefore, extending `AI-Hilbert` to encompass these non-polynomial settings would be of interest.

We point out that several authors have already proposed extensions of the sum-of-squares paradigm beyond polynomial basis functions, and these works offer a promising starting point for performing such an extension. Namely, Löfberg and Parrilo[49] (see also Bach[50] and Bach and Rudi[51]) propose an extension to trigonometric basis functions, and Fawzi et al.[52] propose approximating univariate non-polynomial functions via their Gaussian quadrature and Padé approximants. Moreover, Huchette and Vielma[53] advocate modeling non-convex functions via piecewise linear approximations with strong dual bounds. Using such polynomial approximations of non-polynomial operators offers one promising path for extending `AI-Hilbert` to the non-polynomial setting.

### Automatic parameter-tuning of AI-Hilbert

Our method `AI-Hilbert` requires hyperparameter optimization by the user to trade-off the importance of fidelity to a model, fidelity to experimental data, and complexity of the symbolic model. Therefore, one extension of this work could be to automate this hyperparameter optimization process, by automatically solving mixed-integer and semidefinite optimization problems with different bounds on the degree of the proof certificates and different weights on the relative importance of fidelity to a model and fidelity to data, and using

machine learning techniques to select solutions most likely to satisfy a scientist using `AI-Hilbert`; see also ref. 54 for a review of automated hyperparameter optimization.

## Improving the scalability of AI-Hilbert

One limitation of our implementation of `AI-Hilbert` is that it relies on reformulating sum-of-squares optimization problems as semidefinite problems and solving them via primal-dual interior point methods (IPMs)[17,55]. This arguably presents a limitation because the Newton step in IPMs see, e.g., ref. 56 requires performing a memory-intensive matrix inversion operation. Indeed, this matrix inversion operation is sufficiently expensive that, in our experience, `AI-Hilbert` was unable to perform scientific discovery tasks with more than $n = 15$ variables and a constraint on the degree of the certificates searched over of $d = 20$ or greater (in general, runtime and memory usage is a function of both the number of symbolic variables and the degree of the proof certificates searched over).

To address this limitation and enhance the scalability of `AI-Hilbert`, there are at least three future directions to explore. First, one could exploit ideas related to the Newton polytope (or convex hull of the exponent vectors of a polynomial)[57] to reduce the number of monomials in the sum-of-squares decompositions developed in this paper, as discussed in detail in ref. 34, Chap 3.3.4. Second, one could use presolving techniques such as chordal sparsity[58,59] or partial facial reduction[60,61] to reduce the number of variables in the semidefinite optimization problems that arise from sum-of-squares optimization problems. Third, one could attempt to solve sum-of-squares problems without using computationally expensive interior point methods for semidefinite programs, e.g., by using a Burer-Monteiro factorization approach[62,63] or by optimizing over a second-order cone inner approximation of the positive semidefinite cone[64].

## Methods

We next state the main theoretical results that underpin `AI-Hilbert` and describe the methods used to address the problems studied to validate `AI-Hilbert`.

### Preliminaries

We let non-boldface characters such as $b$ denote scalars, lowercase bold-faced characters such as $x$ denote vectors, uppercase bold-faced characters such as $A$ denote matrices and calligraphic uppercase characters such as $\mathcal{Z}$ denote sets. We let $[n]$ denote the set of indices $\{1, \ldots, n\}$. We let $e$ denote the vector of ones, $0$ denote the vector of all zeros, and $\mathbb{I}$ denote the identity matrix. We let $\|x\|_p$ denote the $p$-norm of a vector $x$ for $p \geq 1$. We let $\mathbb{R}$ denote the real numbers, $\mathcal{S}^n$ denote the cone of $n \times n$ symmetric matrices, and $\mathcal{S}_+^n$ denote the cone of $n \times n$ positive semidefinite matrices.

We also use some notations specific to the sum-of-squares (SOS) optimization literature; see ref. 32 for an introduction to computational algebraic geometry and ref. 34 for a general theory of sum-of-squares and convex algebraic optimization. Specifically, we let $\mathbb{R}[x]_{n,2d}$ denote the ring of real polynomials in the $n$-tuple of variables $x \in \mathbb{R}^n$ of degree $2d$, $P_{n,2d} := \{p \in \mathbb{R}[x]_{n,2d} : p(x) \geq 0 \ \forall x \in \mathbb{R}^n\}$ denote the convex cone of non-negative polynomials in $n$ variables of degree $2d$, and

$$\Sigma[x]_{n,2d} := \left\{ p(x) : \exists q_i, \ldots, q_m \in \mathbb{R}[x]_{n,d}, p(x) = \sum_{i=1}^m q_i^2(x) \right\} \quad (4)$$

denote the cone of sum-of-squares polynomials in $n$ variables of degree $2d$, which can be optimized over via $\binom{n+d}{d}$ dimensional semidefinite matrices (c.f. ref. 16) using interior point methods[17]. Note that $\Sigma[x]_{n,2d} \subseteq P_{n,2d}$, and the inclusion is strict unless $n \leq 2$, $2d \leq 2$ or

$n = 3, 2d = 4$[47]. Nonetheless, $\Sigma[x]_{n,2d}$ provides a high-quality approximation of $P_{n,2d}$, since each non-negative polynomial can be approximated (in the $\ell_1$ norm of its coefficient vector) to any desired accuracy $\epsilon > 0$ by a sequence of sum-of-squares[65]. If the maximum degree $d$ is unknown, we suppress the dependence on $d$ in our notation.

To define a notion of distance between polynomials, we also use several functional norms. Let $\|\cdot\|_p$ denote the $\ell_p$ norm of a vector. Let $\mu \in \mathbb{N}^n$ be the vector $(\mu_1, \ldots, \mu_n)$ and $x^\mu$ stand for the monomial $x_1^{\mu_1} \cdots x_n^{\mu_n}$. Then, for a polynomial $q \in \mathbb{R}[x]_{n,2d}$ with the decomposition $q(x) = \sum_{\mu \in \mathbb{N}^n : \|\mu\|_1 \leq 2d} a_\mu x^\mu$, we let the notation $\|q\|_p = (\sum_{\mu \in \mathbb{N}^n : \|\mu\|_1 \leq 2d} a_\mu^p)^{1/p}$ denote the coefficient norm of the polynomial,

Finally, to derive new laws of nature from existing ones, we repeatedly invoke a fundamental result from real algebraic geometry called the Positivstellensatz (see, e.g., ref. 66). Various versions of the Positivstellensatz exist, with stronger versions holding under stronger assumptions see ref. 67, for a review, and any reasonable version being a viable candidate for our approach. For simplicity, we invoke a compact version due to[33], which holds under some relatively mild assumptions but nonetheless lends itself to relatively tractable optimization problems:

**Theorem 1.** (Putinar's Positivstellensatz[33], see also Theorem 5.1 of ref. 16): Consider the basic (semi) algebraic sets

$$\mathcal{G} := \left\{ x \in \mathbb{R}^n : g_1(x) \geq 0, \ldots, g_k(x) \geq 0 \right\}, \quad (5)$$

$$\mathcal{H} := \left\{ x \in \mathbb{R}^n : h_1(x) = 0, \ldots, h_l(x) = 0 \right\}, \quad (6)$$

where $g_i, h_j \in \mathbb{R}[x]_n$, and $\mathcal{G}$ satisfies the Archimedean property (see also ref. 34, Chap. 6.4.4), i.e., there exists an $R > 0$ and $\alpha_0, \ldots \alpha_k \in \Sigma[x]_n$ such that $R - \sum_{i=1}^n x_i^2 = \alpha_0(x) + \sum_{i=1}^k \alpha_i(x) g_i(x)$.

Then, for any $f \in \mathbb{R}[x]_{n,2d}$, the implication

$$x \in \mathcal{G} \cap \mathcal{H} \Rightarrow f(x) \geq 0 \quad (7)$$

holds if and only if there exist SOS polynomials $\alpha_0, \ldots, \alpha_k \in \Sigma[x]_{n,2d}$, and real polynomials $\beta_1, \ldots, \beta_l \in \mathbb{R}[x]_{n,2d}$ such that

$$f(x) = \alpha_0(x) + \sum_{i=1}^k \alpha_i(x) g_i(x) + \sum_{j=1}^l \beta_j(x) h_j(x). \quad (8)$$

Note that strict polynomial inequalities of the form $h_i(x) > 0$ can be modeled by introducing an auxiliary variable $\tau$ and requiring that $h_i(x)\tau^2 - 1 = 0$, and thus our focus on non-strict inequalities in Theorem 1 is without loss of generality see also ref. 34.

Remarkably, the Positivstellensatz implies that if we set the degree of $\alpha_i$s to be zero, then polynomial laws consistent with a set of equality-constrained polynomials can be searched over via linear optimization. Indeed, this subset of laws is sufficiently expressive that, as we demonstrate in our numerical results, it allows us to recover Kepler's third law and Einstein's dilation law axiomatically. Moreover, the set of polynomial natural laws consistent with polynomial (in)equalities can be searched via semidefinite or sum-of-squares optimization.

### Overall problem setting: combining theory and data

`AI-Hilbert` aims to discover an unknown polynomial model $q(x) = 0$, which contains one or more dependent variables raised to some power within the expression (to avoid the trivial solution $q \equiv 0$), is

approximately consistent with our axioms $\mathcal{G}$ and $\mathcal{H}$ -meaning $d^c$ is small, and explains our experimental data well, meaning $q(\bar{x}_i)$ is small for each data point $i$, and is of low complexity.

Let $x_1, \ldots, x_t$ denote the measurable variables, and let $x_1$ denote the dependent variable which we would like to ensure appears in our scientific law. Let $\Omega = \{\boldsymbol{\mu} \in \mathbb{N}^n : \|\boldsymbol{\mu}\|_1 \leq 2d\}$. Let the discovered polynomial expression by

$$q(x) = \sum_{\boldsymbol{\mu} \in \Omega} a_{\boldsymbol{\mu}} x^{\boldsymbol{\mu}}, \qquad (9)$$

where $2d$ is a bound on the maximum allowable degree of $q$. We formulate the following polynomial optimization problem:

$$\min_{q \in \mathbb{R}_{n,2d}} \sum_{\bar{x}_i \in \mathcal{D}} |q(\bar{x}_i)| + \lambda \cdot d^c(q, \mathcal{G} \cap \mathcal{H})$$
$$\text{s.t.} \qquad \sum_{\boldsymbol{\mu} \in \Omega : \mu_1 \geq 1} a_{\boldsymbol{\mu}} = 1, \qquad (10)$$
$$a_{\boldsymbol{\mu}} = 0 \ \forall \boldsymbol{\mu} \in \Omega : \sum_{j=t+1}^{n} \mu_j \geq 1,$$

where $d^c$, the distance between $q$ and the background theory, is the optimal value of an inner minimization problem, $\lambda > 0$ is a hyperparameter that balances the relative importance of model fidelity to the data against model fidelity to a set of axioms, the first constraint ensures that $x_1$, our dependent variable, appears in $q$, the second constraint ensures that we do not include any unmeasured variables. In certain problem settings, we constrain $d^c = 0$, rather than penalizing the size of $d^c$ in the objective.

Note that the formulation of the first constraint controls the complexity of the scientific discovery problem via the degree of the Positivstellensatz certificate: a smaller bound on the allowable degree in the certificate yields a more tractable optimization problem but a less expressive family of certificates to search over, which ultimately entails a trade-off that needs to be made by the user. Indeed, this trade-off has been formally characterized by Lasserre[65], who showed that every non-negative polynomial is approximable to any desired accuracy by a sequence of sum-of-squares polynomials, with a trade-off between the degree of the SOS polynomial and the quality of the approximation.

After solving Problem (10), one of two possibilities occurs. Either the distance between $q$ and our background information is 0, or the Positivstellensatz provides a non-zero polynomial

$$r(\boldsymbol{x}) := q(\boldsymbol{x}) - \alpha_0(\boldsymbol{x}) - \sum_{i=1}^{k} \alpha_i(\boldsymbol{x}) g_i(\boldsymbol{x}) - \sum_{j=1}^{l} \beta_j(\boldsymbol{x}) h_j(\boldsymbol{x}) \qquad (11)$$

which defines the discrepancy between our derived physical law and its projection onto our background information. In this sense, solving Problem (10) also provides information about the inverse problem of identifying a complete set of axioms that explain $q$. In either case, it follows from the Positivstellensatz that solving Problem (10) for different hyperparameter values and different bounds on the degree of $q$ eventually yields polynomials that explain the experimental data well and are approximately derivable from background theory.

### Discovering scientific laws from background theory alone

Suppose that the background theory $\mathcal{B}$ constitutes a complete set of axioms that fully describe our physical system. Then, any polynomial that contains our dependent variable $x_1$ and is derivable from our system of axioms is a valid physical law. Therefore, we need not even collect any experimental data, and we can solve the following

feasibility problem to discover a valid law (let $\Omega = \{\boldsymbol{\mu} \in \mathbb{N}^n : \|\boldsymbol{\mu}\|_1 \leq 2d\}$):

$$\exists \qquad q(x) = \sum_{\boldsymbol{\mu} \in \Omega} a_{\boldsymbol{\mu}} x^{\boldsymbol{\mu}}$$
$$\text{s.t.} \qquad q(\boldsymbol{x}) = \alpha_0(\boldsymbol{x}) + \sum_{j=1}^{k} \alpha_i(\boldsymbol{x}) g_i(\boldsymbol{x}) + \sum_{j=1}^{l} \beta_j(\boldsymbol{x}) h_j(\boldsymbol{x}),$$
$$\sum_{\boldsymbol{\mu} \in \Omega : \mu_1 \geq 1} a_{\boldsymbol{\mu}} = 1, \qquad (12)$$
$$a_{\boldsymbol{\mu}} = 0 \ \forall \boldsymbol{\mu} \in \Omega : \sum_{j=t+1}^{n} \mu_j \geq 1,$$
$$\alpha_i(\boldsymbol{x}) \in \Sigma[\boldsymbol{x}]_{n,2d}, \beta_j(\boldsymbol{x}) \in \mathbb{R}[\boldsymbol{x}]_{n,2d},$$

where the second and third constraints ensure that we include the dependent variable $x_1$ in our formula $q$ and rule out the trivial solution $q = 0$, and exclude any solutions $q$ that contain uninteresting symbolic variables respectively.

Note that if we do not have any inequality constraints in either problem, then we may often eliminate $\alpha_i$ and obtain a linear optimization problem.

### Hagen-Poiseuille equation

We consider the problem of deriving the velocity of laminar fluid flow through a circular pipe, from a simplified version of the Navier-Stokes equations, an assumption that the velocity can be modeled by a degree-two polynomial in the radius of the pipe, and a no-slip boundary condition. Let $u(r)$ denote the velocity in the pipe where $r$ is the distance from the center of the pipe, $R$ denotes the radius of the pipe, $\Delta p$ denotes the pressure differential throughout the pipe, $L$ denotes the length of the pipe, and $\mu$ denote the viscosity of the fluid, we have the following velocity profile for $r \in [0, R]$:

$$u(r) = \frac{-\Delta p}{4L\mu}(r^2 - R^2). \qquad (13)$$

We now derive this law axiomatically by assuming that the velocity profile can be modeled by a symmetric polynomial and iteratively increasing the degree of the polynomial until we obtain a polynomial solution, consistent with Occam's Razor. Accordingly, we initially set the degree of $u$ to be two and add together the following terms with appropriate polynomial multipliers:

$$u = c_0 + c_2 r^2, \qquad (14)$$

$$\mu \frac{\partial}{\partial r}(r \frac{\partial}{\partial r} u) - r \frac{dp}{dx} = 0, \qquad (15)$$

$$c_0 + c_2 R^2 = 0, \qquad (16)$$

$$L \frac{dp}{dx} = -\Delta p. \qquad (17)$$

Here Equation (14) posits a quadratic velocity profile in $r$, Equation (15) imposes a simplified version of the Navier-Stokes equations in spherical coordinates, Equation (16) imposes a no-slip boundary condition on the velocity profile of the form $u(R) = 0$, and Equation (17) posits that the pressure gradient throughout the pipe is constant. The variables in this axiom system are $u, r, R, L, \mu, \Delta p, c_0, c_2$, and $\frac{dp}{dx}$. We treat $c_0, c_2, \frac{dp}{dx}$ as variables that cannot be measured and use the `differentiate` function in `Julia` to symbolically differentiate $u = c_0 + c_2 r^2$ with respect to $r$ in Equation (15) before solving the problem, giving the equivalent expression $4c_2\mu r - r\frac{dp}{dx}$. Solving Problem (12) with $u$ as the dependent variable, and searching for polynomial multipliers (and polynomial $q$) of degree at most 3 in each variable and an overall

degree of at most 6, we get the expression:

$$4rL\mu u - r\Delta p(R^2 - r^2) = 0, \tag{18}$$

which confirms the result. The associated polynomial multipliers for Equations (14)–(17) are:

$$4rL\mu, \tag{19}$$

$$r^2 L - LR^2, \tag{20}$$

$$4rL\mu, \tag{21}$$

$$r^3 - rR^2. \tag{22}$$

### Radiated gravitational wave power

We now consider the problem of deriving the power radiated from gravitational waves emitted by two-point masses orbiting their common center of gravity in a Keplerian orbit, as originally derived by Peters and Mathews[68] and verified for binary star systems by Hulse and Taylor[69]. Specifically,[68] showed that the average power generated by such a system is:

$$P = -\frac{32G^4}{5c^5r^5}(m_1 m_2)^2(m_1 + m_2), \tag{23}$$

where P is the (average) power of the wave, $G = 6.6743 \times 10^{-11}\,\mathrm{m^3\,kg^{-1}\,s^{-2}}$ is the universal gravitational constant, c is the speed of light, $m_1$, and $m_2$ are the masses of the objects, and we assume that the two objects orbit a constant distance of r away from each other. Note that this equation is one of the twenty so-called bonus laws considered in the work introducing AI-Feynman[22], and notably, is one of only two such laws that neither AI-Feynman nor Eureqa[70] were able to derive. We now derive this law axiomatically, by combining the following axioms with appropriate multipliers:

$$\omega^2 r^3 - G(m_1 + m_2) = 0, \tag{24}$$

$$x5(m_1 + m_2)^2 c^5 P + G\mathrm{Tr}\left(\frac{d^3}{dt^3}\left(m_1 m_2 r^2 \begin{pmatrix} x^2 - \frac{1}{3} & xy & 0 \\ xy & y^2 - \frac{1}{3} & 0 \\ 0 & 0 & -\frac{1}{3} \end{pmatrix}\right)\right)^2 = 0, \tag{25}$$

$$x^2 + y^2 = 1, \tag{26}$$

where we make the variable substitution $x = \cos\phi, y = \sin\phi$, Tr stands for the trace function, and we manually define the derivative of a bivariate degree-two trigonometric polynomial in $\phi = \phi_0 + \omega t$ in $(x, y)$ in terms of $(x, y, \omega)$ as the following linear operator:

$$\frac{d}{dt}\left(\begin{pmatrix} \sin\phi \\ \cos\phi \end{pmatrix}^\top \begin{pmatrix} a_{1,1} & a_{1,2} \\ a_{2,1} & a_{2,2} \end{pmatrix}\begin{pmatrix} \sin\phi \\ \cos\phi \end{pmatrix}\right)$$
$$= \omega\begin{pmatrix} \sin\phi \\ \cos\phi \end{pmatrix}^\top \begin{pmatrix} a_{1,2} + a_{2,1} & a_{1,1} - a_{2,2} \\ a_{1,1} - a_{2,2} & -a_{1,2} - a_{2,1} \end{pmatrix}\begin{pmatrix} \sin\phi \\ \cos\phi \end{pmatrix}. \tag{27}$$

Note that Equation (24) is a restatement of Kepler's previously derived third law of planetary motion, Equation (25) provides the gravitational power of a wave when the wavelength is large compared to the source dimensions, by linearizing the equations of general

relativity, the third equation defines the quadruple moment tensor, and Equation (26) (which we state as $x^2 + y^2 = 1$ within our axioms) is a standard trigonometric identity. Solving Problem (12) with P as the dependent variable, and searching for a formula involving $P, G, r, c, m_1, m_2$ with polynomial multipliers of degree at most 20, and allowing each variable to be raised to power for the variables $(P, x, y, \omega, G, r, c, m_1, m_2)$ of at most $(1, 4, 4, 4, 3, 6, 1, 5, 5)$ respectively, then yields the following equation:

$$\frac{1}{4}Pr^5 c^5(m_1 + m_2)^2 = \frac{-8}{5}G^4 m_1^2 m_2^2(m_1 + m_2)^3, \tag{28}$$

which verifies the result. Note that this equation is somewhat expensive to derive, owing to the fact that searching over the set of degree 20 polynomial multipliers necessitates generating a large number of linear equalities, and writing these equalities to memory is both time and memory-intensive. Accordingly, we solved Problem (12) using the MIT SuperCloud environment[71] with 640 GB RAM. The resulting system of linear inequalities involves 416392 candidate monomials, and takes 14368s to write the problem to memory and 6.58s to be solved by `Mosek`. This shows that the correctness of the universal gravitational wave equation can be confirmed via the following multipliers:

$$\frac{-8}{5}Gm_1^2 m_2^2\left(\omega^4 r^6(x^2 + y^2)^2 + \omega^2 r^3 G(m_1 + m_2) + G^2(m_1 + m_2)^2\right), \tag{29}$$

$$\frac{1}{20}r^5, \tag{30}$$

$$\frac{-8}{5}\omega^4 r^6 G^2 m_1^2 m_2^2(m_1 + m_2)(x^2 + y^2 + 1). \tag{31}$$

Finally, Fig. 4 illustrates how the Positivstellensatz derives this equation, by demonstrating that (setting $m_1 = m_2 = c = G = 1$), the gravitational wave equation is precisely the set of points $(\omega, r, P)$ where our axioms hold with equality.

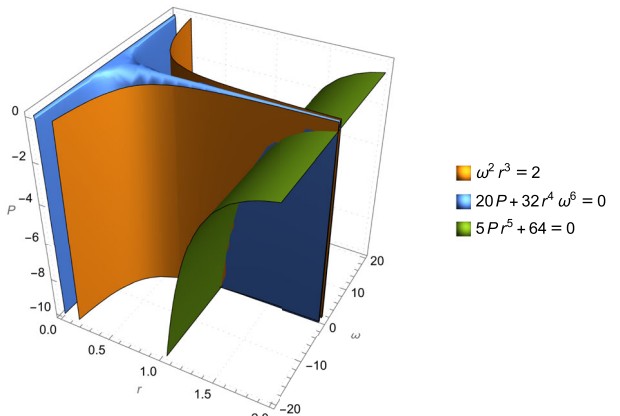

**Fig. 4 | Intersection of background theory axioms for the Radiation Gravitational Wave Power Equation.** Illustration of the Positivstellensatz and its ability to recover the Radiation Gravitational Wave Power Equation in the special case where $m_1 = m_2 = c = G = 1$. Keeping other variables constant, the points that obey the power equation are the intersection of the points that obey Kepler's Third Law and the points of a linearized equation from general relativity, and the wave equation is recoverable by adding these other equations with appropriate polynomial multipliers.

## Einstein's Relativistic Time Dilation Law

Next, we consider the problem of deriving Einstein's relativistic time dilation formula from a complete set of background knowledge axioms plus an inconsistent Newtonian axiom, which posits that light behaves like a mechanical object. We distinguish between these axioms using data on the relationship between the velocity of a light clock and the relative passage of time, as measured experimentally by Chou et al.[72] and stated explicitly in the work of Cornelio et al.[43], Tab. 6.

Einstein's law describes the relationship between how two observers in relative motion to each other observe time and demonstrates that observers moving at different speeds experience time differently. Indeed, letting the constant $c$ denote the speed of light, the frequency $f$ of a clock moving at a speed $v$ is related to the frequency $f_0$ of a stationary clock via

$$\frac{f}{f_0} = \sqrt{1 - \frac{v^2}{c^2}}. \tag{32}$$

We now derive this law axiomatically, by adding together the following five axioms with appropriate polynomial multipliers:

$$cdt_0 - 2d = 0, \tag{33}$$

$$cdt - 2L = 0, \tag{34}$$

$$4L^2 + 4d^2 - v^2 dt^2 = 0, \tag{35}$$

$$f dt_0 = 1, \tag{36}$$

$$f dt = 1, \tag{37}$$

plus the following (inconsistent) Newtonian axiom:

$$dt^2(v^2 + c^2) - 4L^2 = 0, \tag{38}$$

where $dt_0$ denotes the time required for a light to travel between two stationary mirrors separated by a distance $d$, and $dt$ denotes the time required for light to travel between two similar mirrors moving at velocity $v$, giving a distance between the mirrors of $L$.

These axioms have the following meaning: Equation (33) relates the time required for light to travel between two stationary mirrors to their distance, Equation (34) similarly relates the time required for light to travel between two mirrors in motion to the effective distance $L$, Equation (35) relates the physical distance between the mirrors $d$ to their effective distance $L$ induced by the motion $v$ via the Pythagorean theorem, and Equations (36), (37) relate frequencies and periods. Finally, Equation (38) assumes (incorrectly) that light behaves like other mechanical objects, meaning if it is emitted orthogonally from an object traveling at velocity $v$, then it has velocity $\sqrt{v^2 + c^2}$.

By solving Problem (10) with a cardinality constraint c.f. refs. 73,74 that we include at most $\tau = 5$ axioms (corresponding to the exclusion of one axiom), a constraint that we must exclude either Equation (34) or Equation (38), $f$ as the dependent variable, experimental data in $f, f_0, v, c$ to separate the valid and invalid axioms (obtained from[43], Tab. 6 by setting $f_0 = 1$ to transform the data in $(f - f_0)/f_0$ into data in $f, f_0$), $f_0, v, c$ as variables that we would like to appear in our polynomial formula $q(\boldsymbol{x}) = 0 \ \forall \boldsymbol{x} \in \mathcal{G} \cap \mathcal{H}$, and searching the set of polynomial multipliers of degree at most 2 in each term, we obtain the law:

$$-c^2 f_0^2 + c^2 f^2 + f_0^2 v^2 = 0, \tag{39}$$

in 6.04 seconds using `Gurobi` version 9.5.1. Moreover, we immediately recognize this as a restatement of Einstein's law (32). This shows

that the correctness of Einstein's law can be verified by multiplying the (consistent relativistic set of) axioms by the following polynomials:

$$2df_0^2 f^2 + cf_0 f^2, \tag{40}$$

$$-cf_0^2 f - 2f_0^2 f^2 L, \tag{41}$$

$$-f_0^2 f^2, \tag{42}$$

$$-2cdf_0 f^2 - c^2 f^2, \tag{43}$$

$$c^2 dt f_0^2 f - dt f_0^2 f v^2 + c^2 f_0^2 - f_0^2 v^2. \tag{44}$$

Moreover, it verifies that relativistic axioms, particularly the axiom $cdt = 2L$, fit the light clock data of ref. 72 better than Newtonian axioms, because, by the definition of Problem (10), `AI-Hilbert` selects the combination of $\tau = 5$ axioms with the lowest discrepancy between the discovered scientific formula and the experimental data.

## Kepler's Third Law of Planetary Motion

We now consider the problem of deriving Kepler's third law of planetary motion from a complete set of background knowledge axioms plus an incorrect candidate formula as an additional axiom, which is to be screened out using experimental data. To our knowledge, this paper is the first work that addresses this particularly challenging problem setting. Indeed, none of the approaches to scientific discovery reviewed in the introduction successfully distinguish between correct and incorrect axioms via experimental data by solving a single optimization problem. The primary motivation for this experiment is to demonstrate that `AI-Hilbert` provides a system for determining whether, given a background theory and experimental data, it is possible to improve upon a state-of-the-art scientific formula using background theory and experimental data.

Kepler's law describes the relationship between the distance between two bodies, e.g., the sun and a planet, and their orbital periods and takes the form:

$$p = \sqrt{\frac{4\pi^2 (d_1 + d_2)^3}{G(m_1 + m_2)}}, \tag{45}$$

where $G = 6.6743 \times 10^{-11} \ m^3 \ kg^{-1} \ s^{-2}$ is the universal gravitational constant, $m_1$, and $m_2$ are the masses of the two bodies, $d_1$ and $d_2$ are the respective distances between $m_1, m_2$ and their common center of mass, and $p$ is the orbital period. We now derive this law axiomatically by adding together the following five axioms with appropriate polynomial multipliers:

$$d_1 m_1 - d_2 m_2 = 0, \tag{46}$$

$$(d_1 + d_2)^2 F_g - Gm_1 m_2 = 0, \tag{47}$$

$$F_c - m_2 d_2 w^2 = 0, \tag{48}$$

$$F_c - F_g = 0, \tag{49}$$

$$wp = 1, \tag{50}$$

plus the following (incorrect) candidate formula (as an additional axiom) proposed by Cornelio et al.[43] for the exoplanet dataset (where

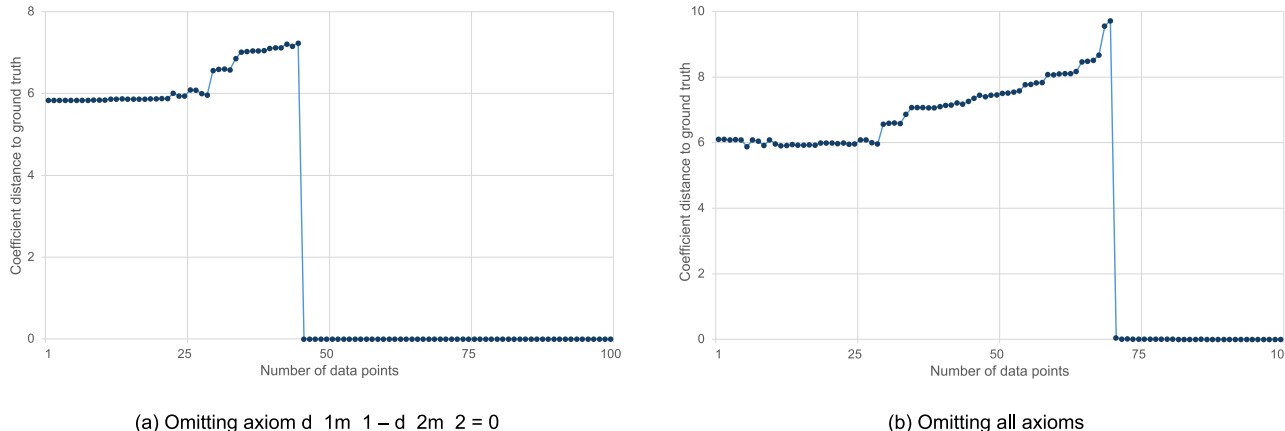

**Fig. 5 | Coefficient distance between scientific formula derived by AI-Hilbert and ground truth.** The *x*-axis depicts the number of data points where we omit some axioms (**a**), or all axioms (**b**).

the mass of the planets can be discarded as negligible when added to the much bigger mass of the star):

$$p^2 m_1 - 0.1319(d_1 + d_2)^3 = 0 . \tag{51}$$

Here $F_g$ and $F_c$ denote the gravitational and centrifugal forces in the system, and $w$ denotes the frequency of revolution. Note that we replaced $p$ with $2\pi p$ in our definition of revolution period in order that $\pi$ does not feature in our equations; we divide $p$ by $2\pi$ after deriving Kepler's law.

The above axioms have the following meaning: Equation (46) defines the center of mass of the dynamical system, Equation (47) defines the gravitational force of the system, Equation (48) defines the centrifugal force of the system, Equation (49) matches the centrifugal and dynamical forces, and Equation (50) relates the frequency and the period of revolution.

Accordingly, we solve our polynomial optimization problem under a sparsity constraint that at most $\tau = 5$ axioms can be used to derive our model, a constraint that $d^c = 0$ (meaning we need not specify the hyperparameter $\lambda$ in (10)), by minimizing the objective

$$\sum_{i=1}^{n} |q(\bar{\boldsymbol{x}}_i)|, \tag{52}$$

where $q$ is our implicit polynomial and $\{\bar{\boldsymbol{x}}_i\}_{i=1}^{4}$ is a set of observations of the revolution period of binary stars stated in ref. 43, Tab. 5. Searching over the set of degree-five polynomials $q$ derivable using degree six certificates then yields a mixed-integer linear optimization problem in 18958 continuous and 6 discrete variables, with the solution:

$$m_1 m_2 G p^2 - m_1 d_1 d_2^2 - m_2 d_1^2 d_2 - 2 m_2 d_1 d_2^2 = 0, \tag{53}$$

which is precisely Kepler's third law. The validity of this equation can be verified by adding together our axioms with the weights:

$$-d_2^2 p^2 w^2, \tag{54}$$

$$-p^2, \tag{55}$$

$$d_1^2 p^2 + 2 d_1 d_2 p^2 + d_2^2 p^2, \tag{56}$$

$$d_1^2 p^2 + 2 d_1 d_2 p^2 + d_2^2 p^2, \tag{57}$$

$$m_1 d_1 d_2^2 p w + m_2 d_1^2 d_2 p w + 2 m_2 d_1 d_2^2 p w + m_1 d_1 d_2^2 + m_2 d_1^2 d_2 + 2 m_2 d_1 d_2^2, \tag{58}$$

as previously summarized in Fig. 3. This is significant, because existing works on symbolic regression and scientific discovery[22,75] often struggle to derive Kepler's law, even given observational data. Indeed, our approach is also more scalable than deriving Kepler's law manually; Johannes Kepler spent four years laboriously analyzing stellar data to arrive at his law[76].

### Kepler's law revisited with missing axioms
We now revisit the problem of deriving Kepler's third law of planetary motion considered, with a view to verifying AI-Hilbert's ability to discover scientific laws from a combination of theory and data. Specifically, rather than providing a complete (albeit inconsistent) set of background theories, we suppress a subset of the axioms (46)–(50) and investigate the number of noiseless data points required to recover Equation (53). To simplify our analysis, we set $G = 1$ and generate noiseless data observations by sampling the values of the independent variables (the masses of the two bodies and the distance between them) uniformly at random in the ranges observed in real data (i.e., exoplanet dataset in AI-Descartes[43]) and computing the value of the dependent variable (the revolution period) using the ground truth formula.

To exploit the fact that our data observations are noiseless, we solve the following variant of (10):

$$\min_{q \in \mathbb{R}_{n,2d}} \quad \frac{\lambda_1}{\sqrt{|\mathcal{D}|}} \sum_{\bar{\boldsymbol{x}}_i \in \mathcal{D}} |q(\bar{\boldsymbol{x}}_i)| + \lambda_2 \cdot d^c(q, \mathcal{G} \cap \mathcal{H}) + (1 - \lambda_1 - \lambda_2) \parallel q \parallel_1$$

$$\text{s.t.} \quad \sum_{\boldsymbol{\mu} \in \Omega : \boldsymbol{\mu}_1 \geq 1} a_{\boldsymbol{\mu}} = 1,$$

$$a_{\boldsymbol{\mu}} = 0 \;\; \forall \boldsymbol{\mu} \in \Omega : \sum_{j=t+1}^{n} \boldsymbol{\mu}_j \geq 1,$$

$$\sum_{\bar{\boldsymbol{x}}_i \in \mathcal{D}} |q(\bar{\boldsymbol{x}}_i)| \leq |\mathcal{D}|\epsilon,$$

$$\sum_{\boldsymbol{\mu} \in \Omega : \boldsymbol{\mu}_1 = 0} a_{\boldsymbol{\mu}} \leq -1/10 \bigvee \sum_{\boldsymbol{\mu} \in \Omega : \boldsymbol{\mu}_1 = 0} a_{\boldsymbol{\mu}} \geq 1/10$$

$$\tag{59}$$

where we set $\lambda_1 = 0.9, \lambda_2 = 0.01, \epsilon = 10^{-7}$ for all experiments, seek a degree 4 polynomial $q$ using a proof certificate of degree at most 6, and use $L_1$-coefficient norm of $q$ as a regularization term analogously to Lasso regression[77]. Note that the second-to-last constraint ensures that the derived polynomial $q$ explains all (noiseless) observations up to a small tolerance. Further, the last constraint is imposed as a linear

inequality constraint with an auxiliary binary variable via the big-$M$ technique[78,79], to ensure that the derived formula includes at least one term not involving the rotation period.

Figure 5 depicts the $\ell_2$ coefficient distance between the scientific formula derived by AI-Hilbert and Equation (53) mod $d_1 m_1 - d_2 m_2 = 0$ as we increase the number of data points, where we suppress the axiom $d_1 m_1 - d_2 m_2 = 0$ (left), where we suppress all axioms (right). In both cases, there is an all-or-nothing phase transition [(c.f. ref. 80), for a discussion of this phenomenon throughout machine learning] in AI-Hilbert's ability to recover the scientific law, where before a threshold number of data points, AI-Hilbert cannot recover Kepler's law, and beyond the threshold, AI-Hilbert recovers the law exactly.

Note that the increase in the coefficient distance before $m = 71$ data points reflects that solutions near $q = 0$ (and thus closer in coefficient norm to the ground truth) are optimal with $m = 0$ data points but do not fit a small number of data points perfectly, while polynomials $q$ further from the ground truth in coefficient norm fit a small number of data points perfectly. Indeed, the total error with respect to the training data is less than $10^{-5}$ for all values of $m$ in both problem settings.

Figure 5 reveals that when only the axiom $d_1 m_1 - d_2 m_2 = 0$ is missing, it is possible to perform scientific discovery with as few as 46 data points, while at least 71 data points are needed when all axioms are missing. This is because the axiom $d_1 m_1 - d_2 m_2 = 0$ multiplied by the term in the proof certificate $-d_2^2 p^2 w^2$ is of a similar complexity as Kepler's Third Law. Thus, the value of $d_1 m_1 - d_2 m_2 = 0$ is 46 data points, while the value of all axioms is 71 data points. The value of data compared to background theory depends, in general, on the problem setting and the data quality, as well as how well dispersed the data samples are.

### Bell inequalities

We now consider the problem of deriving Bell Inequalities in quantum mechanics. Bell Inequalities[81] are well-known in physics because they provide bounds on the correlation of measurements in any multi-particle system which obeys local realism (i.e., for which a joint probability distribution exists), that are violated experimentally, thus demonstrating that the natural world does not obey local realism. For ease of exposition, we prove a version called the GHZ inequality[82]. Namely, given random variables $A, B, C$ which take values on $\{\pm 1\}$, for any joint probability distribution describing $A, B, C$, it follows that

$$\mathbb{P}(A = B) + \mathbb{P}(A = C) + \mathbb{P}(B = C) \geq 1, \tag{60}$$

but this bound is violated experimentally[83].

We derive this result axiomatically, using Kolmogorov's probability axioms. In particular, letting $p_{-1,1,-1} = \mathbb{P}(A = -1, B = 1, C = -1)$, deriving the largest lower bound for which this inequality holds is equivalent to solving the following linear optimization problem:

$$\min p_{AB} + p_{BC} + p_{AC} \text{ s.t. } p \in \mathcal{S}, \tag{61}$$

where $\mathcal{S} := \{p \geq \mathbf{0}, e^\top p = 1\}$, $p_{AB} := p_{-1,-1,-1} + p_{-1,-1,1} + p_{1,1,-1} + p_{1,1,1}$ and $p_{AC}, p_{BC}$ are defined similarly.

We solve this problem using Gurobi and Julia, which verifies that $\gamma = 1$ is the largest value for which this inequality holds, and obtains the desired inequality. Moreover, the solution to its dual problem yields the certificate $2p_{-1,-1,-1} + 2p_{1,1,1} \geq 0$, which verifies that 1 is indeed a valid lower bound for $p_{AB} + p_{BC} + p_{AC}$, by adding $e^\top p$ to the left-hand side of this certificate and 1 to the right-hand side.

To further demonstrate the generality and utility of our approach, we now derive a more challenging Bell inequality, namely the so-called I3322 inequality (c.f. ref. 84). Given particles $A_1, A_2, A_3, B_1, B_2, B_3$ which take values on $\{\pm 1\}$, the inequality reveals that for any joint

probability distribution, we have:

$$\begin{aligned}
\mathbb{E}[A_1] - \mathbb{E}[A_2] + \mathbb{E}[B_1] - \mathbb{E}[B_2] - \mathbb{E}[(A_1 - A_2)(B_1 - B_2)] \\
+ \mathbb{E}[(A_1 + A_2)B_3] + \mathbb{E}[A_3(B_1 + B_2)] \leq 4.
\end{aligned} \tag{62}$$

Using the same approach as previously, and defining $p$ to be an arbitrary discrete probability measure on $\{\pm 1\}^6$, we verify that the smallest such upper bound which holds for each joint probability measure is 4, with the following polynomial certificate modulo $e^\top p = 1$:

$$\begin{aligned}
& 4p_{2,1,1,1,1,1} + 4p_{1,2,1,1,1,1} + 8p_{2,2,1,1,1,1} + 4p_{2,1,2,1,1,1} + 4p_{1,2,2,1,1,1} + 8p_{2,2,2,1,1,1} \\
& + 4p_{1,1,1,2,1,1} + 8p_{2,1,1,2,1,1} + 4p_{2,1,2,1,2,1,1} + 8p_{2,2,1,2,1,1} + 4p_{2,1,2,2,1,1} + 4p_{2,2,2,2,1,1} \\
& + 4p_{1,1,1,1,2,1} + 4p_{2,1,1,1,2,1} + 12p_{2,1,1,2,1} + 12p_{2,2,1,1,2,1} + 8p_{1,2,1,1,2,1} + 8p_{2,2,1,1,2,1} \\
& + 8p_{1,1,1,2,2,1} + 8p_{2,1,1,2,2,1} + 12p_{1,2,1,2,2,1} + 12p_{2,1,2,2,1} + 4p_{1,2,2,2,2,1} + 4p_{2,2,2,2,2,1} \\
& + 4p_{1,1,2,1,1,2} + 4p_{2,1,2,1,1,2} + 4p_{1,2,2,1,1,2} + 4p_{2,2,2,1,1,2} + 4p_{1,1,1,2,1,2} + 4p_{2,1,1,2,1,2} \\
& + 4p_{1,1,2,2,1,2} + 4p_{2,1,2,2,1,2} + 4p_{1,1,1,1,2,2} + 8p_{1,2,1,1,2,2} + 4p_{2,1,1,2,2} + 4p_{1,1,2,1,2,2} \\
& + 8p_{1,2,2,1,2,2} + 4p_{2,2,2,1,2,2} + 8p_{1,1,1,2,2,2} + 4p_{2,1,1,2,2,2} + 8p_{1,2,1,2,2,2} + 4p_{2,2,1,2,2,2} \\
& + 4p_{1,1,2,2,2,2} + 4p_{1,2,2,2,2,2} \geq 0
\end{aligned} \tag{63}$$

where an index of 1 denotes that a random variable took the value $-1$ and an index of 2 denotes that a random variable took the value 1, and the random variables are indexed in the order $A_1, A_2, A_3, B_1, B_2, B_3$.

## Data availability
The data used in this study are available in the AI-Hilbert GitHub repository[85]: https://github.com/IBM/AI-Hilbert.

## Code availability
The code used for this work can be found, freely available, at the AI-Hilbert GitHub repository[85]: https://github.com/IBM/AI-Hilbert.

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

## Acknowledgements

We are grateful to Ken Clarkson, Joao Goncalves, Phokion Kolatis, Jon Lenchner, and Nimrod Megiddo (IBM) for valuable discussions on scientific discovery. We acknowledge the Imperial College London Open Access Fund for paying the Article Processing Fee. Ryan Cory-Wright and Bachir El Khadir gratefully acknowledge IBM Research for hosting them as IBM Goldstine postdoctoral fellows while part of this work was conducted.

## Author contributions

R. C. conceptualized the overall scientific discovery framework, designed the project, conceived and designed the experiments, analyzed results to validate the framework, designed and performed the experiments, and co-authored the manuscript. C. C. contributed to the conceptualization of the discovery framework, performed some of the experiments, generated the data, formatted code and data for the release, co-authored the manuscript, and designed the figures. S. D. contributed to the conceptualization of the discovery framework, designed and performed some of the experiments, analyzed results to validate the framework, and co-authored the manuscript. B. E. K. conceptualized the overall scientific scientific discovery framework, helped design the background theory distance metric, and performed initial proof-of-concept experimentation. L. H. contributed to the conceptualization of the discovery framework, designed and performed some of the experiments, analyzed results to validate the framework, and co-authored the manuscript.

## Competing interests

The authors declare no competing interests.
