## [Peer Review File · Nature Communications]

REVIEWER COMMENTS

Reviewer #1 (Remarks to the Author):

The paper addresses an important problem in science: deducing universal laws from observations. The authors provide a comprehensive literature review on the subject. The proposed approach brings together distinct criteria for learning (sum of squares and polynomial optimization) in an innovative interaction that permits efficient discovery of constrained relationships. AI-Hilbert is proposed as an extension of the scientific method to enable autonomous discovery. This capability will arguably allow for scientific discoveries using just the right amount of correct data. The idea is reasonable with logical scrutiny and with more detailed mathematical scrutiny. The proposed approach relies on very interesting (and for some strange reason, unexplored) properties of positive polynomials, developed by Putinar and Lasserre over many years.

The paper was a pleasure to read, is written for readers with diverse backgrounds in science and engineering. The topic of the paper is already very timely and likely to become more so.

The paper is recommended for publication as is.

Reviewer #2 (Remarks to the Author):

This article focusses on the timely topic of automatic knowledge discovery, in particular, learning of physical laws. The authors propose a novel approach, which uses theoretical information in terms of axioms as well as experimental data in a very flexible manner, which also sets it apart from the closest previous approach coined AI-Descartes. The approach is validated in several numerical experiments.

In general, this is an interesting approach, but if considered for a leading journal such as Nature Communications, the paper would need to be significantly improved. The following items are of major concern to me:

- * The approach is explained much too vague. To judge it properly and also for implementing it, a detailed step-by-step description is imperative.

- * Table 1 compares the different approaches in just two problem settings. For a leading scientific article, I would expect a much more detailed comparison. In fact, I would expect the authors to not just take the experiments from other work, but to use their code or implement it themselves to produce a more comprehensive comparison.

- * Noisy data is a major concern for almost all "learning physical law" approaches. This is not sufficiently detailed in this paper. In fact, in the numerical experiments (from Section 2.5 on), noise is barely mentioned. To produce insightful experiments in this regard, for each of the equations (unless they are solely derived from axioms, which is anyhow the not-that-interesting case) I expect the authors to test their approach in various noise settings, validating the robustness of AI Hilbert.

- * It is interesting to discuss the balance between "background theory" and "experimental data". However, not only is this balance not described on a high level (just roughly in the experimental section), but it is also missing what type of "experimental data" is sufficient for success, etc.

- * The three Examples (Example 1 to 3) on a first glance look like results by the authors, whereas there is nothing new and it is merely theory taken from other work. I would at least expect some additional theoretical insights, preferably on the noise stability of the approach.

Response to Reviewer 1

Summary and Recommendation

The paper addresses an important problem in science: deducing universal laws from observations. The authors provide a comprehensive literature review on the subject. The proposed approach brings together distinct criteria for learning (sum of squares and polynomial optimization) in an innovative interaction that permits efficient discovery of constrained relationships. AI-Hilbert is proposed as an extension of the scientific method to enable autonomous discovery. This capability will arguably allow for scientific discoveries using just the right amount of correct data. The idea is reasonable with logical scrutiny and with more detailed mathematical scrutiny. The proposed approach relies on very interesting (and for some strange reason, unexplored) properties of positive polynomials, developed by Putinar and Lasserre over many years.

The paper was a pleasure to read, is written for readers with diverse backgrounds in science and engineering. The topic of the paper is already very timely and likely to become more so.

The paper is recommended for publication as is.

Thank you very much for the detailed and encouraging review and the positive appreciation of our work; we were delighted to read your review. We likewise agree that it is unusual that the polynomial optimization community has not yet explored scientific discovery, and we very much hope that the ideas contained in the manuscript prove of use to the scientific community in accelerating scientific discovery.

Response to Reviewer 2

Summary and Recommendation

This article focuses on the timely topic of automatic knowledge discovery, in particular, learning of physical laws. The authors propose a novel approach, which uses theoretical information in terms of axioms as well as experimental data in a very flexible manner, which also sets it apart from the closest previous approach coined AI-Descartes. The approach is validated in several numerical experiments.

In general, this is an interesting approach, but if considered for a leading journal such as Nature Communications, the paper would need to be significantly improved. The following items are of major concern to me:

Thank you for your detailed and constructive summary of the paper, and for the time and effort you spent reviewing our work. We have revised our manuscript to address your comments and respond to each of them below.

In particular, we have added a comparison with existing scientific discovery methods on a test set of scientific discovery problems (please see Appendix A of the revised manuscript), twelve new examples of scientific discovery using the approach (please see Appendix B of the revised manuscript), and made some other additions and modifications to the paper which, in our view, improved it; please find these summarized on the last page of this letter.

We hope this resolves your major concerns about our submission, and look forward to your feedback.

Major Comment 1

The approach is explained much too vague. To judge it properly and also for implementing it, a detailed step-by-step description is imperative.

As part of the revision, we have added a new section to the paper (Section 2.1), which provides detailed overview of our approach, including defining the inputs to and outputs from the approach and pseudocode describing our approach, another new section (Section 3.1) which fully describes the lower-level implementation of AI-Hilbert on a stylized example, including writing out the linear optimization problem that arises from this scientific discovery setting, and illustrating how the optimal solution varies as we vary the number of axioms and amount of data available. Further, we are in the process of open-sourcing our code at ai-hilbert.github.io which will be available soon, illustrating how our method works in practice and allowing it to be implemented by anyone with a working installation of the Julia programming language and a mixed-integer conic optimization solver such as Gurobi or CPLEX.

We believe that by reading the revised manuscript in conjunction with our open-sourced code, it should be possible for the reader to apply our approach to new scientific discovery settings.

Major Comment 2

* Table 1 compares the different approaches in just two problem settings. For a leading scientific article, I would expect a much more detailed comparison. In fact, I would expect the authors to not just take the experiments from other work, but to use their code or implement it themselves to produce a more comprehensive comparison.

Thanks. In Appendix A of the revised manuscript, we now compare AI-Hilbert against four widely used existing methods from the literature (namely, AI-Descartes, AI-Feynman, PySR, and BMS) on a suite of 14

different examples (Table 1), using the code from these other works. We identify that, because AI Hilbert can integrate data with theory, it can recover scientific laws even in noisy settings where other state-of-the-art methods that do not integrate background theory cannot make scientific discoveries.

We would, however, like to clarify that we originally only compared AI-Hilbert on two problems because it is a fundamentally different method to most scientific discovery methods in the literature, and thus hard to compare fairly. Namely, AI-Hilbert integrates background theory with data, while other methods (to our knowledge, with the exception of AI-Descartes, which typically requires at least some experimental data) are data-driven methods that cannot integrate background theory. Thus, of the five problem settings we considered in the original system, we only put two in our original Table 1, because the remaining three did not have any experimental data and thus could not have been recovered by other methods. However, for the new experiments we generated synthetic data (resembling real data – with noise and few data points) to perform this comparison. The generation follows standard practice, similarly to what has been done in AI-Descartes and AI-Feynman (more details in Appendix A).

Further, we would like to highlight that entries in Table 1 denoted with a \checkmark^* indicate a limitation of data-driven methods. Namely, because methods from the literature are fully data-driven, they cannot exactly derive constants associated with scientific laws. Instead, they usually group information concerning multiple constants into one that is less interpretable. For instance, consider the Compton Scattering problem (appendix B.7), the correct formula is $l_2 = l_1 + \frac{h}{m_e c}(1 - \cos(\theta))$ while a data-driven method can usually only derive $l_2 = l_1 + k(1 - \cos(\theta))$ (where k is a real number). Our method, by incorporating data directly into the background theory, can provide a fine grained expression for the derived formula that includes the expression $\frac{h}{m_e c}$.

We hope this satisfies your concerns about the number of problem settings in our manuscript.

Major Comment 3

* Noisy data is a major concern for almost all "learning physical law" approaches. This is not sufficiently detailed in this paper. In fact, in the numerical experiments (from Section 2.5 on), noise is barely mentioned. To produce insightful experiments in this regard, for each of the equations (unless they are solely derived from axioms, which is anyhow the not-that-interesting case) I expect the authors to test their approach in various noise settings, validating the robustness of AI Hilbert.

Thanks. We agree that noise is a major concern when learning physical laws, especially when a method is benchmarked exclusively using synthetic data. Accordingly, we thought about including additional experiments that explore the performance of AI-Hilbert as we reduce or increase the amount of noise in our data generation processes. However, we ultimately decided to not include them, because, in this work, we considered problem settings with real data, which is more realistic than noisy synthetic data. Specifically, in Sections 3.5 and 3.6, we demonstrate using real data from experiments that our approach successfully recovers the relativistic time dilation law and Kepler's third law of planetary motion.

Major Comment 4

* It is interesting to discuss the balance between "background theory" and "experimental data". However, not only is this balance not described on a high level (just roughly in the experimental section), but it is also missing what type of "experimental data" is sufficient for success, etc.

Thank you for raising this important point. We agree that the balance between background theory is a very interesting question to explore. Accordingly, we have made the following changes to the paper to contribute to the community's understanding of the trade-off between data and theory:

- We have edited section 2.4 (impact of background theory on amount of data needed to discover scientific laws) to clarify that a partial answer to this question is given by the statistics community, who show that imposing relevant constraints on a machine learning model reduces the amount of data required to recover a ground truth model, when one exists.
- We have added a new section (section 3.2) which discusses the trade-off between data and theory at a more fundamental level, and points out that more background theory implies that less data is needed to make scientific discoveries because background theory reduces the effective VC-dimension of the scientific discovery process, and thus the existence of this theory vs. data trade-off essentially follows from the machine learning literature on generalization. This section also provides a geometric interpretation of an axiom as an infinite amount of data in a particular subspace, which (we hope) sheds further light on this matter.
- We have added another new section (section 3.7) which revisits the Kepler example given access to noiseless data and when either the first axiom or all axioms are missing, and demonstrates that less data (45 data points) is required to recover Kepler's law when one axiom is missing than when all axioms are missing (71 data points).

We hope that these additional contributions satisfy your concerns about the extent to which we study the trade-off between background theory and data.

Major Comment 5

* The three Examples (Example 1 to 3) on a first glance look like results by the authors, whereas there is nothing new and it is merely theory taken from other work. I would at least expect some additional theoretical insights, preferably on the noise stability of the approach.

Thank you for pointing out that the contributions in section 2.4 needed to be made clearer. We agree (and intended to write section 2.4 in a way that makes clear that) the examples on sparse linear regression/sparse PCA/matrix completion in Section 2.4 summarize results already known in the literature; we have edited the first paragraph of Section 2.4 to make this clearer and apologize for any confusion caused.

The purpose of section 2.4 is to highlight that machine learning is a special case of scientific discovery where it is well documented that introducing relevant background theory (e.g., by restricting the complexity of the derived law) can reduce the amount of data required to obtain a valid scientific law. Thus, background theory reduces the data needed to make discoveries in at least some discovery settings.

We considered analyzing these examples in more detail and deriving new results based on sparse linear regression/sparse PCA etc. which account for noise in the data generation process. However, both examples 1 and 2 already relate the amount of data required to perform scientific discovery to the amount of noise in the data generation process (with the parameters σ and λ , respectively). Moreover, it is also worth noting that even deriving another result like this would likely require a separate paper in its own right: for instance, in example 2, the first couple of sentences we summarized from Arous et al. (2020) required the majority of a 63 page COLT paper to prove. Therefore, we believe that additional theoretical insights of this form are beyond the scope of this work.

Summary of Other Changes Made to the Paper:

In addition to the changes requested by both referees and the senior editor, we have made the following changes to the manuscript, broken down into major changes and minor changes:

Major:

- We are in the process of open-sourcing our code on GitHub at `ai-hilbert.github.io` to ensure the reproducibility of our work.

Minor:

- We have updated our schematic illustration of AI Hilbert (Figure 3) and its caption, to provide an overview of the system in general terms.
- We have clarified why we called our approach AI-Hilbert at the beginning of section 1.2.
- We have added a paragraph to section 1.3 of the paper clarifying that strict inequalities can be modeled using equalities and an auxiliary variable, thus modeled using our approach. Further, we added a paragraph clarifying that we could use an alternative version of the Pstatz with only inequality constraints, and explaining why this would be less computationally tractable.
- We added a paragraph to section 2.3 of the paper pointing out that our representation of scientific formulae as implicit polynomials introduces some degeneracy in the set of optimal polynomials derivable via our approach, and pointed out that our proposed approach of constraining the degree of the proof certificate (partially) breaks this degeneracy.
- We have added another 18 citations throughout the paper, to enhance our literature review and justify some of the statements made throughout the paper.

References

G. B. Arous, A. S. Wein, and I. Zadik. Free energy wells and overlap gap property in sparse PCA. In *Conference on Learning Theory*, pages 479–482. PMLR, 2020.

REVIEWERS' COMMENTS

Reviewer #2 (Remarks to the Author):

The authors addressed my concerns, the paper is significantly improved, and I am happy to recommend publication.

Reviewer #2 (Remarks on code availability):

The authors addressed my concerns, the paper is significantly improved, and I am happy to recommend publication.